# Risk of adverse events in gastrointestinal endoscopy: Zero-inflated Poisson regression mixture model for count data and multinomial logit model for the type of event

**Marco Gemma**[1]*, **Fulvia Pennoni**[2], **Roberta Tritto**[2], **Massimo Agostoni**[3]

**1** Anesthesia & Intensive Care, Fatebenefratelli Hospital, Milan, Italy, **2** Department of Statistics and Quantitative Methods, University of Milano-Bicocca, Milan, Italy, **3** Anesthesia & Intensive Care, S. Raffaele Hospital, Milano, Italy

* marco.gemma@asst-fbf-sacco.it

**Data Availability Statement:** Data contain potentially identifying or sensitive patient information. The data underlying the results

## Abstract

### Background and aims

We analyze the possible predictive variables for Adverse Events (AEs) during sedation for gastrointestinal (GI) endoscopy.

### Methods

We consider 23,788 GI endoscopies under sedation on adults between 2012 and 2019. A Zero-Inflated Poisson Regression Mixture (ZIPRM) model for count data with concomitant variables is applied, accounting for unobserved heterogeneity and evaluating the risks of multi-drug sedation. A multinomial logit model is also estimated to evaluate cardiovascular, respiratory, hemorrhagic, other AEs and stopping the procedure risk factors.

### Results

In 7.55% of cases, one or more AEs occurred, most frequently cardiovascular (3.26%) or respiratory (2.77%). Our ZIPRM model identifies one population for non-zero counts. The AE-group reveals that age >75 years yields 46% more AEs than age <66 years; Body Mass Index (BMI) ≥27 27% more AEs than BMI <21; emergency 11% more AEs than routine. Any one-point increment in the American Society of Anesthesiologists (ASA) score and the Mallampati score determines respectively a 42% and a 16% increment in AEs; every hour prolonging endoscopy increases AEs by 41%.

Regarding sedation with propofol alone (the sedative of choice), adding opioids to propofol increases AEs by 43% and adding benzodiazepines by 51%.

Cardiovascular AEs are increased by age, ASA score, smoke, in-hospital, procedure duration, midazolam/fentanyl associated with propofol.

Respiratory AEs are increased by BMI, ASA and Mallampati scores, emergency, in-hospital, procedure duration, midazolam/fentanyl associated with propofol.

presented in the study are available from the Ethics Committee of the Ospedale S. Raffaele - Scientific Inst. S. Raffaele - Via Olgettina, 60 - 20132 Milano. Director Dr. Michela Franzin <franzin.michela@hsr.it> Responsabile Segreteria Tecnico Scientifica del CE - Ospedale San Raffaele - Dibit 2 - San Michele 2 - piano 6° - via Olgettina 60 - 20132 Milano.

**Funding:** The authors received no specific funding for this work.

**Competing interests:** The authors have declared that no competing interests exist.

Hemorrhagic AEs are increased by age, in-hospital, procedure duration, midazolam/fentanyl associated with propofol.

The risk of suspension of the endoscopic procedure before accomplishment is increased by female gender, ASA and Mallampati scores, and in-hospital, and it is reduced by emergency and procedure duration.

## Conclusions

Age, BMI, ASA score, Mallampati score, in-hospital, procedure duration, other sedatives with propofol increase the risk for AEs during sedation for GI endoscopy.

## Introduction

Sedation is often needed to improve the tolerability of gastrointestinal (GI) endoscopy procedures. In this setting, a crucial issue is to identify predictive factors for sedation-related adverse events (AEs) [1–3]. Besides adding to anesthesiological knowledge, this interest also stems from the current debate about possible non-anesthesiologist-delivered sedation in GI. Although previous studies effectively addressed this topic [4], some significant issues require further reporting. First of all, the population of patients undergoing GI endoscopy changes over time, and endoscopic procedures are more often performed on older and more compromised patients [5], especially in countries suffering from demographic declines, like Italy. Moreover, operative—and not simply diagnostic—procedures are more frequently performed in GI [6].

Another issue that deserves attention is that analytical methods adequate for count data have been much improved recently. Due to the low occurrence of AEs, we employ a Zero-Inflated Poisson Regression Mixture (ZIPRM) model [7] with concomitant variables to model counts of AEs. It is a model-based clustering method that allows us to accommodate unobserved heterogeneity in the population [8]. In this way, we account for a possible source of overdispersion, occurring when there is high variability in the data regarding the expected mean due to the excess of zeros and or unmeasured confounders. The Zero-Inflated Poisson (ZIP) model [9–11] is often used to handle zero-inflated count data when the inflation is not excessive and thus permitting to account for this source of overdispersion. The zero counts are supposed to arise from a Poisson distribution (also defined as "sampling zeros") and from a Bernoulli distribution (also defined as "structural zeros"). The zero-inflated Negative Binomial model is also used to account for overdispersion [12]. More recently, the ZIPRM model was useful to model a variety of count data since the heterogeneity across individuals is tackled by assuming that the mean event rate has a discrete distribution with a finite number of components (or classes) and assuming that within each class, it depends on some covariates (risk factors). The estimation of the model parameters is based on the maximization of the log-likelihood function through the Expectation-Maximization algorithm [13], and it is performed by alternating two steps until convergence. The number of components is selected by relying on the Bayesian Information Criterion [14].

In this work we report a significant Italian University Hospital experience with a high volume of activity. We focus on 23,788 sedations for GI endoscopy performed between 2012 and March 2019. We propose the ZIPRM model to evaluate continuous and categorical risk factors for AEs, including all the different drugs that were used for sedation (opioids, benzodiazepines, curare, halogenated gases, propofol, and others). Since propofol is the most frequently used

sedative and is often associated with the other drugs, we built a second ZIPRM model to evaluate whether this association affects the occurrence of AEs with respect to pure propofol sedation. In this way, we addressed the occurrence of AEs of any kind. We also propose a multinomial logit model [15] to disentangle the risk factors for the main types of AEs, namely cardiovascular, respiratory, hemorrhagic, and for stopping the endoscopic procedure before its accomplishment. The remainder of the paper is organized as follows. In Section Materials and Methods, we describe available data. In Section Statistical Models, we illustrate the proposed models. In Section Results, first we show the estimated parameters for the ZIPRM model comparing all the drugs. Second, we show the ZIPRM model results when we consider as drug propofol alone or propofol combined with other drugs. We then show the results concerning the estimated multinomial logit model and compare the odds ratios for cardiovascular, respiratory and hemorrhagic events. In the Discussion section, we provide some conclusions.

## Materials and methods

The study was approved by the Ethics Committee of the Fondazione Centro S. Raffaele del Monte Tabor—Scientific Inst. S. Raffaele—Milano (CE 07/04/2011) and all patients signed an informed consent form for GI endoscopy under sedation that explicitly mentioned data collection and their use for anonymous scientific reporting.

GI endoscopies performed in adult patients ($> 18$ years old) under sedation between January 2012 and March 2019 at the University hospital S. Raffaele in Milano (Italy) are considered. Data were prospectively collected in a specific database for clinical and scientific purposes. The need for sedation was determined in agreement between the patient, the endoscopist, and the anesthesiologist during the routine preoperative clinical evaluation. The endoscopic maneuvers were performed by experienced certified gastroenterologists (a total of 13 physicians over the study period) assisted by trained nurses. Sedation was performed by senior anesthesiologists (a total of 30 physicians over the study period) or by in-specialty-training anesthesiologists supervised by them (30 trainees changing every 6 months).

Monitoring and sedative administration were performed as described in Agostoni et al. [4]. In particular, propofol was administered with a target-controlled infusion (TCI) (Diprifusor, Pilote Anesthesie IS; Fresenius Vial SA, Brézins, France or Terufusion-TIVA/TCI TE372 Terumo Europe N.V., Leuven, Belgium). The pharmacokinetic model used was the Marsh model. After a starting dose (target 4–5 µg/mL for patients younger than 80 years and with an American Society of Anesthesiologists (ASA) score 1 or 2; target 2–3 µg/mL in the other patients), the maintenance dose was adjusted to achieve adequate sedation. Midazolam, the only benzodiazepine we used, was administered in intravenous boluses to achieve adequate sedation. Fentanyl was administered in intravenous boluses on an on-demand basis according to the anesthesiologist's judgment when analgesia was needed. Pharyngeal anesthesia was never used. The halogenated gas sevoflurane was administered by inhalation. If needed, muscle relaxation was achieved with intravenous rocuronium bromide 0.6 mg/Kg, and general anesthesia was induced with intravenous propofol 2 mg/Kg and maintained with inhaled sevoflurane.

The attending anesthesiologists filled in a dedicated database recording the following variables: sex, age, body mass index (BMI), smoking habits, American Society of Anesthesiologists (ASA) score, Mallampati score, duration of the procedure, need to stop the procedure before its accomplishment, procedure type (routine versus emergent), procedure purpose (diagnostic versus procedural), and type of sedative drug used.

The ASA score is a physical status classification system widely used to assess and communicate a patient's pre-anesthesia medical co-morbidities [15]. There are six possible ASA

categories: 1 normal health, 2 mild systemic disease, 3 severe systemic disease, 4 systemic disease with the constant threat to life, 5 moribund state, and 6 brain death. Although this was not an *a priori* exclusion criterion, none of our patients exhibited an ASA status 5 or 6.

The modified Mallampati score [16] is commonly used to predict tracheal intubation difficulty and is assessed by asking the patient, in a sitting position, to open his/her mouth and protrude the tongue as much as possible. According to the anatomical structures visualized through the oral cavity, the Mallampati score can take the following values: 1 soft palate, uvula, fauces, and pillars visible, 2 soft palate, major part of uvula, and fauces visible, 3 soft palate and base of uvula visible, or 4 only hard palate visible.

AEs occurring during sedation were recorded if they required some anesthesiologist's intervention, such as drug or fluid administration, cardiopulmonary resuscitation, bag-mask ventilation, or tracheal intubation (simple jaw-thrust or chin-lift were not considered). The decision to undertake such interventions was left to the attending senior anesthesiologist. AEs were meant to be unfavorable events related to sedation and seriously impairing baseline patients' conditions. A classification of the AEs in three major categories was adopted, namely cardiovascular (such as arrhythmias, cardiac arrest, bradycardia, electromechanical dissociation, hypertension, shock), respiratory (such as SpO2<90%, respiratory arrest, upper airway obstruction, aspiration in the lungs, respiratory failure), and hemorrhagic. Other AEs (such as nausea and vomiting, hypoglycemia, viscus perforation) were also registered together with the stopping of the endoscopic procedure before its accomplishment.

## Statistical analysis

We use flexible regression models for counts with high inflation of zeros to detect if there is unobserved heterogeneity among patients and to investigate the role played by the explanatory variables. First, we employ the ZIPRM models with concomitant variables (8). After that, we employ a multinomial logit model [17] with covariates to better disentangle risk factors within each category of AEs.

## Zero-inflated Poisson mixture regression model

Poisson regression models [18,19] are often used to handle count data in practical situations; however, in its basic version, this model postulates the homogeneity of the underlying population, and the assumption of equal mean and variance is often restrictive in many situations. In the applied context such as that of the proposed application, the ZIPRM model can be used to investigate the heterogeneity of the population that is, if there are latent subgroups of patients sharing common features accounting for the fact that counts of AEs have an excess of zeroes derived from the data generative mechanism. The ZIPRM model considers a binary distribution degenerated at zero to account for the zero occurrences and a mixture of ordinary Poisson distributions to model counts, see, among others, Alfò and Trovato [20]. In this way, we can account for the heterogeneity among patients, which cannot be explained based on observable patient covariates. The model classifying patients according to the observed response and covariates into homogeneous subpopulations that are internally cohesive and well separated from one another, whose number is determined by the estimation procedure.

We consider $Y_i$ as the non-negative integer count of the AEs for patient $i$, $i = 1,\ldots,n$, with $y$ being a generic element of $Y$. Under the assumption that distribution of counts is zero inflated, we model the possible source of unobserved heterogeneity by considering the probability of observing any specific count as a mixture of ZIP models having a finite unknown number of

components ranging from 1 to $K$. We consider

$$P(Y = y) = \begin{cases} \pi_1 + \pi_2 e^{-\lambda_2} + \cdots + \pi_k e^{-\lambda_k}, & \text{if } y = 0, \\ \pi_2 \dfrac{e^{-\lambda_2}\lambda_2{}^y}{y!} + \cdots + \pi_k \dfrac{e^{-\lambda_k}\lambda_k{}^y}{y!}, & \text{if } y > 0, \end{cases} \tag{1}$$

where $\pi_k$ is the weight of the $k$-th component, and $\pi_1$ is the weight related to the proportion of excess zeros, under the constraints that $0 < \pi_k < 1$ for $k = 1, \ldots, K$, and $\sum_{k=1}^{K} \pi_k = 1$. When $K = 2$ the above model reduces to a ZIP model [21,22].

In the following, we assume the $\log(\lambda_{i,k})$ to be a linear function of the covariates through intercept and regression parameters which are specific for each component. We denote $X_i$ as the vector of baseline patient characteristics with $x_i$ denoting a realization. The model can be formulated as follows

$$P(Y_i = y_i | x_i) = \pi_1 I_{y_i=0} + \sum_{k=2}^{K} \pi_k Pois(y_i | \lambda_{i,k}(x)), \ i = 1, \ldots, n,$$

where $I_{(.)}$ is equal to 1 if the specified condition is met and 0 otherwise, and $Pois(.)$ denotes the Poisson probability mass function of $y_i$ with expected mean $\lambda_{i,k}(x)$, which has the following expression:

$$\log[\lambda_{i,k}(x)] = \beta_{i0,k} + x_i'\beta_{i1,k}, \ i = 1, \ldots, n, \ k = 2, \ldots, K.$$

In order to estimate the model parameters collected into the vector $\theta$, based on a sample of $n$ independent patients for which we observe the counts of the occurrences and covariates, $y_i$ we rely on the log-likelihood function defined as follows

$$l(\theta) = \sum_{i=1}^{n} \log p(y_i | x_i).$$

This function is maximized through iterative algorithms such as those described in [23], in particular we employ the Expectation-Maximization [EM] algorithm [13,24]. The EM algorithm allows us to solve, in an indirect way, the system of likelihood equations by considering the complete data log-likelihood, that is, the likelihood we could compute if knew the assignment of each patient to each component. The algorithm alternates the following two steps: E-step, through which by assigning initial values for the parameters, we compute the conditional expected value of the *complete data log-likelihood*; M-step, through which we maximize this expected value of the vector of parameters, and we update the estimated values of the parameters. The two steps are iterated until suitable convergence criteria are reached. Proper strategies to initialize the model parameters are applied, and search strategies are implemented to explore the parameter space in order to overcome the problem of the multi-modality of the log-likelihood function.

Furthermore, to choose the suitable number of components, when this number is not known a priori, some model selection criteria, such as that proposed in [25], can be applied, and similarly they can be employed to choose the relevant covariates to include in the regression for the expected value of the counts. We use a criterion named Bayesian Information Criterion (BIC), which is based on an index obtained by simply modifying the maximum log-likelihood function at convergence to consider the complexity of the model and defined as

$$\text{BIC}_k = -2l(\hat{\theta}) + \log(n) \cdot \#par,$$

where $l(\hat{\theta})$ denotes the maximum of the log-likelihood when the model has $k$ components, *#par* the number of free parameters, and $n$ the sample size. According to this index, among

different models, the one with the lowest BIC value is preferred. There are other goodness-of-fit criteria, which can be employed for model selection, such as the Akaike Information Criterion [26], that adds as penalty term only $2 \cdot \#par$ or the Integrated Completed Likelihood (ICL) [27], which adds a measure of the uncertainty to the BIC related to the allocation of the units to the clusters. Each unit is allocated to a cluster according to the estimated posterior probability, and the EM algorithm directly provides this probability. For a review of the model selection procedures, see, among others, McLachlan and Peel [8]. Different criteria may point to select a different model, and in this case, other considerations such as the number of components allocated in each cluster and the interpretability of the components should be considered. Standard errors for the parameter estimates are obtained as the square root of the diagonal elements of the inverse of the observed or expected information matrix computed through numerical methods.

## Multinomial log-linear model

In order to evaluate the effects of the baseline covariates on the different typologies of AEs (cardiovascular, respiratory, hemorrhagic, others) and for stopping the procedure before its accomplishment, we consider the polytomous response variable $Z_i$ with $l$ response categories denoted by $z = 0,\ldots, l-1$ for each patient $i$, $i = 1,\ldots,n$. The conditional probability that a patient $i$ manifests AEs of type $z$ depends on the vector of covariates $\boldsymbol{x}_i$ available for each patient, and it is denoted as $P(Z_i = z|\boldsymbol{x}_i)$. Such dependence is formulated according to a multinomial logit parameterization as follows:

$$\log \frac{P(Z_i = z|\boldsymbol{x}_i)}{P(Z_i = 0|\boldsymbol{x}_i)} = \alpha_z + \boldsymbol{x}_i'\boldsymbol{\beta}_z, z = 1,\ldots, l-1,$$

where $\alpha_z$ is an intercept specific of the response category, and $\boldsymbol{\beta}_z$ the vector whose elements denote the individual covariate's effect on the logit of $Z_i = z$ with respect to $Z_i = 0$. Since the baseline category represents the absence of AEs, a positive or negative value of the regression parameter denotes an increase or a decrease in moving from the reference category to the chosen category.

## Results

During the eight-year study period, 74,570 GI endoscopic procedures were performed in the S. Raffaele University Hospital, one of Milano's leading hospitals (Italy). Sedation was required in 24,608 (32.99%) of these procedures and was always administered by anesthesiologists. Since 753 (3.05%) of these procedures were performed in patients less than 18 years old and 67 (0.27%) had incomplete records, 23,788 endoscopic procedures are considered in the present study.

These endoscopic procedures were: 9164 (38.50%) ultrasound endoscopies, 4894 (20.60%) esophago-gastro-duodenoscopies, 2756 (11.60%) endoscopic retrograde colangio-pancreatographies, 2364 (6.90%) colonoscopies, 703 (3.00%) endoscopic mucosal resections, 498 (2.10%) percutaneous endoscopic gastrostomies, 492 (2.10%) esophageal, enteral or colonic prosthesis, 372 (1.60%) endoscopic hemostasis, 169 (0.70%) variceal banding, and 2377 (9.90%) other operative endoscopies.

The proposed models are estimated by using these data through the open source software R [28] and the packages flexmix (Flexible mixture modeling, [29]) for the ZIPRM model and nnet (Fit Neural Network [30]) for the multinomial log-linear model. The complete results are available from the authors upon request, and the code to estimate the proposed model is

available from the Github repository at the following link https://github.com/penful/Gastro_Endos.

Table 1 reports some descriptive statistics of the available data. Propofol was the most frequently used sedative agent (23,289 cases 97.90%), and in 19,795 cases (83.21%) it was administered as a single hypnotic agent while in 3,494 cases (14.69%), it was variably associated with midazolam and or fentanyl. In 249 (1.05%) cases, muscle relaxation was required, and general anesthesia was administered. In 1,797 (7.55%) cases, one or more than one AEs occurred. The most frequent AEs were cardiovascular (776 cases, 3.26%) or respiratory (660 cases, 2.77%). Hemorrhages occurred in 22 cases (0.09%). Other AEs occurred in 95 cases (0.39%). Endoscopy had to be suspended before accomplishment in 244 cases (1.03%).

Table 2 shows the differences between patients sedated only with propofol and patients sedated with propofol associated with midazolam and or fentanyl. Table 3 reports the counts of AEs for these two groups of patients.

**Table 1. Baseline demographic and clinical characteristics.**

| Covariates | Categories | Value | Percentage |
|---|---|---|---|
| Age (years) | < 66 | 11,064 | 46.51 |
| | 67–75 | 6,264 | 26.33 |
| | ≥ 75 | 6,460 | 27.16 |
| Average age (s.d.) | 64 (15.02) | | |
| BMI | < 21 | 5,017 | 21.09 |
| | 21–24 | 6,443 | 27.09 |
| | 25–27 | 6,295 | 26.46 |
| | ≥ 27 | 6,033 | 25.36 |
| ASA score | I | 2,710 | 11.39 |
| | II | 13,529 | 56.87 |
| | III | 6,960 | 29.26 |
| | IV | 589 | 2.48 |
| Mallampati score | I | 10,145 | 42.65 |
| | II | 11,042 | 46.42 |
| | III | 2,259 | 9.50 |
| | IV | 342 | 1.44 |
| Smoking habit | Previous smoker | 4,771 | 20.06 |
| | Non-smoker | 15,116 | 63.54 |
| | Smoker | 3,901 | 16.40 |
| Emergency | Election | 22,101 | 92.91 |
| | Emergency | 1,687 | 7.09 |
| Hospitalization | Outpatient | 7,478 | 31.44 |
| | In-hospital | 16,310 | 65.56 |
| Average duration (s.d.) | 43 (31) | | |
| Drugs | Fentanyl | 2,368 | 9.95 |
| | Sevoflurane | 458 | 1.93 |
| | Midazolam | 2,330 | 9.79 |
| | Other | 57 | 0.24 |
| | Curare | 249 | 1.05 |
| | Propofol | 23,289 | 97.90 |

Body Mass Index (BMI) and age categories are defined according to the observed quantiles, standard deviation (s.d.), the average duration of the procedure is in minutes.

**Table 2. Baseline demographic and clinical characteristics concerning the use of propofol or multi-drug sedation.**

| Covariates | Categories | Propofol alone | | Propofol and midazolam/fentanyl | |
|---|---|---|---|---|---|
| | | Value | Percentage | Value | Percentage |
| Age (year) | < 66 | 9,796 | 49.49 | 1,761 | 50.40 |
| | 67–75 | 4,799 | 24.24 | 737 | 21.09 |
| | ≥75 | 5,200 | 26.27 | 996 | 28.51 |
| BMI | < 21 | 4,118 | 20.80 | 776 | 22.21 |
| | 21–24 | 5,414 | 27.35 | 915 | 26.19 |
| | 25–27 | 5,294 | 26.74 | 871 | 24.93 |
| | ≥ 27 | 4,969 | 25.10 | 932 | 26.67 |
| ASA score | I | 2,421 | 12.23 | 281 | 8.04 |
| | II | 11,876 | 59.99 | 1,575 | 45.08 |
| | III | 5,218 | 26.36 | 1,467 | 41.99 |
| | IV | 280 | 1.41 | 171 | 4.89 |
| Mallampati score | I | 8,648 | 43.69 | 1,374 | 39.32 |
| | II | 9,106 | 46.00 | 1,683 | 48.17 |
| | III | 1,781 | 9.00 | 380 | 10.88 |
| | IV | 260 | 1.31 | 57 | 1.63 |
| Smoking habit | Previous smoker | 3,851 | 17.94 | 786 | 22.5 |
| | Non-smoker | 12,643 | 63.87 | 2158 | 61.76 |
| | Smoker | 3,301 | 16.68 | 550 | 15.74 |
| Emergency | Election | 18,740 | 94.67 | 3,071 | 87.89 |
| | Emergency | 1,055 | 5.33 | 423 | 12.11 |
| Average duration (s.d.) | | 35 (29) | | 40 (40) | |
| Hospitalization | Outpatient | 6,693 | 33.81 | 769 | 22.01 |
| | In-hospital | 13,102 | 66.19 | 2,725 | 77.99 |

Body Mass Index (BMI) and age categories are defined according to the observed quantiles, standard deviation (s.d.), the average duration of the procedure is in minutes.

## Results for Model 1: Zero inflated Poisson mixture regression model

We disentangle the effects of the risk factors by applying a ZIPRM model to the whole patients' observed counts. For this purpose, 23,788 cases are available, the overall sample mean of AEs is 0.038, and the sample variance is 0.044. In order to check for overdispersion, that is when there is high variability in the data with respect to the expected mean [31,32], we estimated a Poisson model by including all the available continuous and categorical covariates, and we performed the test proposed by Cameron and Trivedi [18, Section 3.4] that is based on an auxiliary ordinary least square regression on the fitted values. The test statistic leads to rejecting the hypothesis of equidispersion at a significant level of 99%. Therefore, we estimated the ZIPRM

**Table 3. Number and percentages of adverse events concerning the use of propofol and multi-drug sedation.**

| | Propofol alone | | Propofol and midazolam/fentanyl | |
|---|---|---|---|---|
| Number of adverse events | Value | Percentage | Value | Percentage |
| 0 | 19,262 | 97.31 | 3,227 | 92.36 |
| 1 | 519 | 2.62 | 203 | 5.81 |
| 2 | 14 | 0.07 | 59 | 1.69 |
| 3 | 0 | 0.00 | 5 | 0.14 |

model as a holistic approach to cater to the unexplained heterogeneity and extra dispersion arising in the data, probably due to excess of zeros.

The BIC criterion favored a model with two components, among models estimated with components ranging from 1 to 4. The log-likelihood value at convergence for the chosen model having 25 free parameters is equal to -3,786. The selected model reduces to a ZIP model where the first cluster is referred to the population of patients with no risk of AEs and the second cluster to the population at risk. According to the estimated posterior probabilities, many patients (86%) result in the first cluster. In Fig 1, we show the estimated incidence rate ratios (IRRs) with the confidence intervals calculated at a confidence level of 95%. From these results, we notice that several variables produce a significant increase in AEs. Patients older than 75 years show 44% more AEs than patients younger than 66 years, all the other variables remaining fixed. Patients with a BMI greater or equal to 27 yield an estimated number of AEs 25% greater than that of patients with a BMI less than 21. Concerning the ASA score equal to 1, a score equal to 2 exhibits an 89% increase in AEs, a score equal to 3 or 4 a 153% and 254% increment in AEs, respectively. Concerning a Mallampati score equal to 1, a score equal to 3 exhibits a 43% increment in AEs. In-hospital procedures are 59% more prone to AEs than out-patient procedures. Prolonging endoscopy increases the number of AEs by 59% for every hour. The use of several drugs is associated with more AEs: in particular, fentanyl and midazolam are associated with AEs increases of 40% and 164%, respectively. We better explore this fact with the results presented in the next section concerning comparing patients treated with unique or multiple drugs.

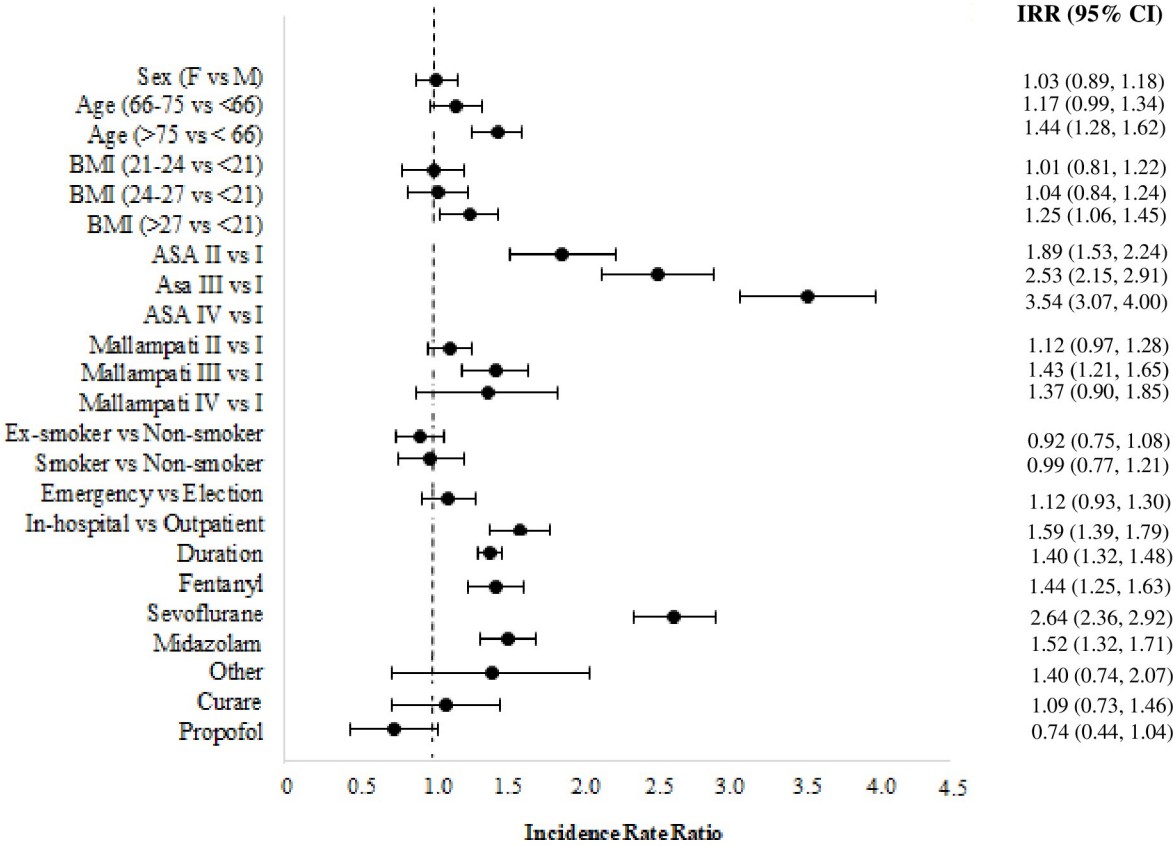

**IRR (95% CI)**

| | |
|---|---|
| Sex (F vs M) | 1.03 (0.89, 1.18) |
| Age (66-75 vs <66) | 1.17 (0.99, 1.34) |
| Age (>75 vs < 66) | 1.44 (1.28, 1.62) |
| BMI (21-24 vs <21) | 1.01 (0.81, 1.22) |
| BMI (24-27 vs <21) | 1.04 (0.84, 1.24) |
| BMI (>27 vs <21) | 1.25 (1.06, 1.45) |
| ASA II vs I | 1.89 (1.53, 2.24) |
| Asa III vs I | 2.53 (2.15, 2.91) |
| ASA IV vs I | 3.54 (3.07, 4.00) |
| Mallampati II vs I | 1.12 (0.97, 1.28) |
| Mallampati III vs I | 1.43 (1.21, 1.65) |
| Mallampati IV vs I | 1.37 (0.90, 1.85) |
| Ex-smoker vs Non-smoker | 0.92 (0.75, 1.08) |
| Smoker vs Non-smoker | 0.99 (0.77, 1.21) |
| Emergency vs Election | 1.12 (0.93, 1.30) |
| In-hospital vs Outpatient | 1.59 (1.39, 1.79) |
| Duration | 1.40 (1.32, 1.48) |
| Fentanyl | 1.44 (1.25, 1.63) |
| Sevoflurane | 2.64 (2.36, 2.92) |
| Midazolam | 1.52 (1.32, 1.71) |
| Other | 1.40 (0.74, 2.07) |
| Curare | 1.09 (0.73, 1.46) |
| Propofol | 0.74 (0.44, 1.04) |

**Fig 1. Estimated incidence rate ratio of the ZIPRM model (Model 1) for the number of adverse events in endoscopy, confidence intervals are provided at a confidence level of 0.95.**

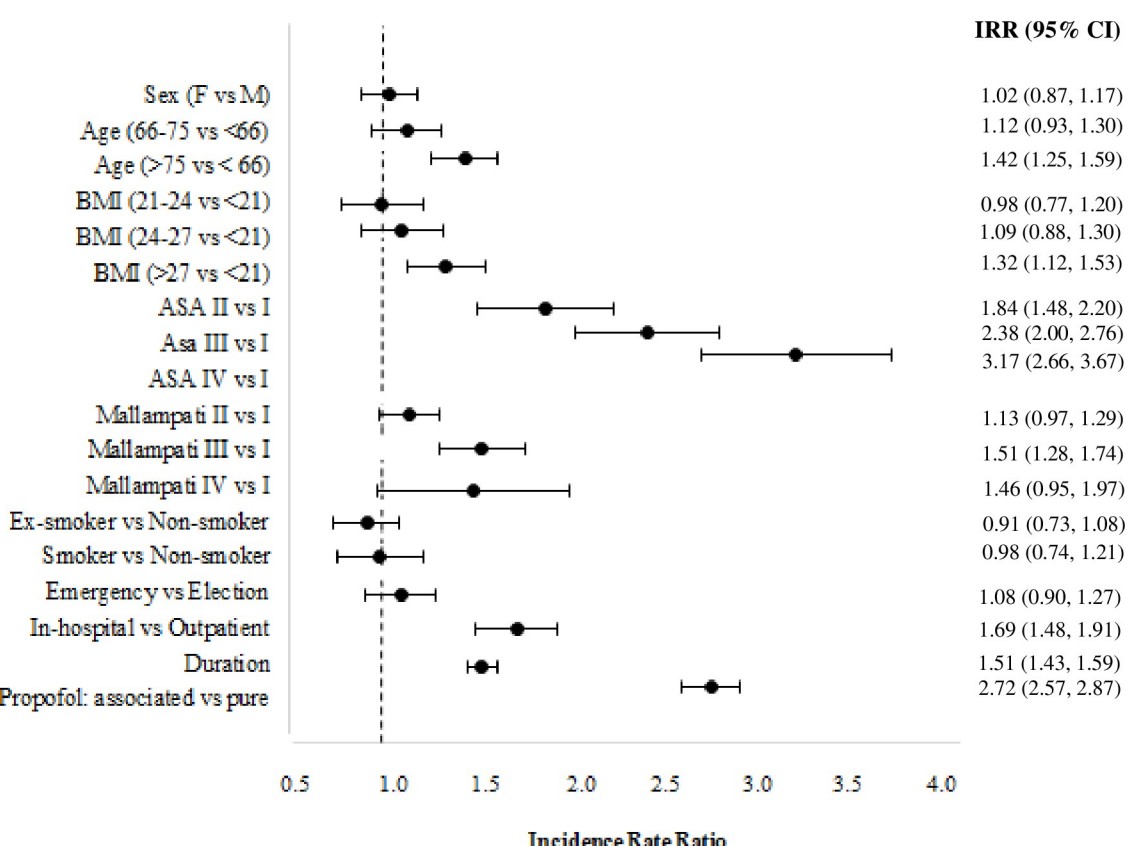

**Fig 2. Estimated incidence rate ratio of the ZIPRM model (Model 2) for the number of adverse events in endoscopy for patients treated with pure propofol or sedated with propofol and other drug, intervals are provided at a confidence level of 0.95.**

## Results of Model 2

We aimed to disentangle the covariate effects considering only those procedures in which propofol was administered alone or associated with other sedatives. We estimated the ZIPRM models using the observations referred to 23,289 patients. The model with two-components was selected through the BIC index from the estimated models with a number of components ranging from 1 to 4. The log-likelihood value at convergence is equal to 3,525 with 20 free parameters. The two components are well-separated: a small proportion of patients (0.07) is assigned to the second cluster corresponding to the population at risk of AEs. The estimated effects for this population are reported in Fig 2 as IRRs with the estimated confidence intervals at a 95% confidence level. Age, BMI, ASA and Mallampati scores, in-hospital procedures, and the procedure's duration behaves similarly as in Model 1. We notice that a clear effect of multi-drug sedation is apparent since patients who received other midazolam/fentanyl in addition to propofol had 172% more AEs than patients who received propofol alone.

## Results of the multinomial logit model

A multinomial logit model (Model 3) is estimated to account for the types of AEs (classified as cardiovascular, respiratory, hemorrhagic, and others) and for stopping the procedure before its accomplishment. For this purpose, a sample of 28,496 patients is considered since some patients may report more than one AE. Fig 3 shows the estimated odds ratios with their asymptotic confidence intervals at a 95% confidence level for cardiovascular events, category

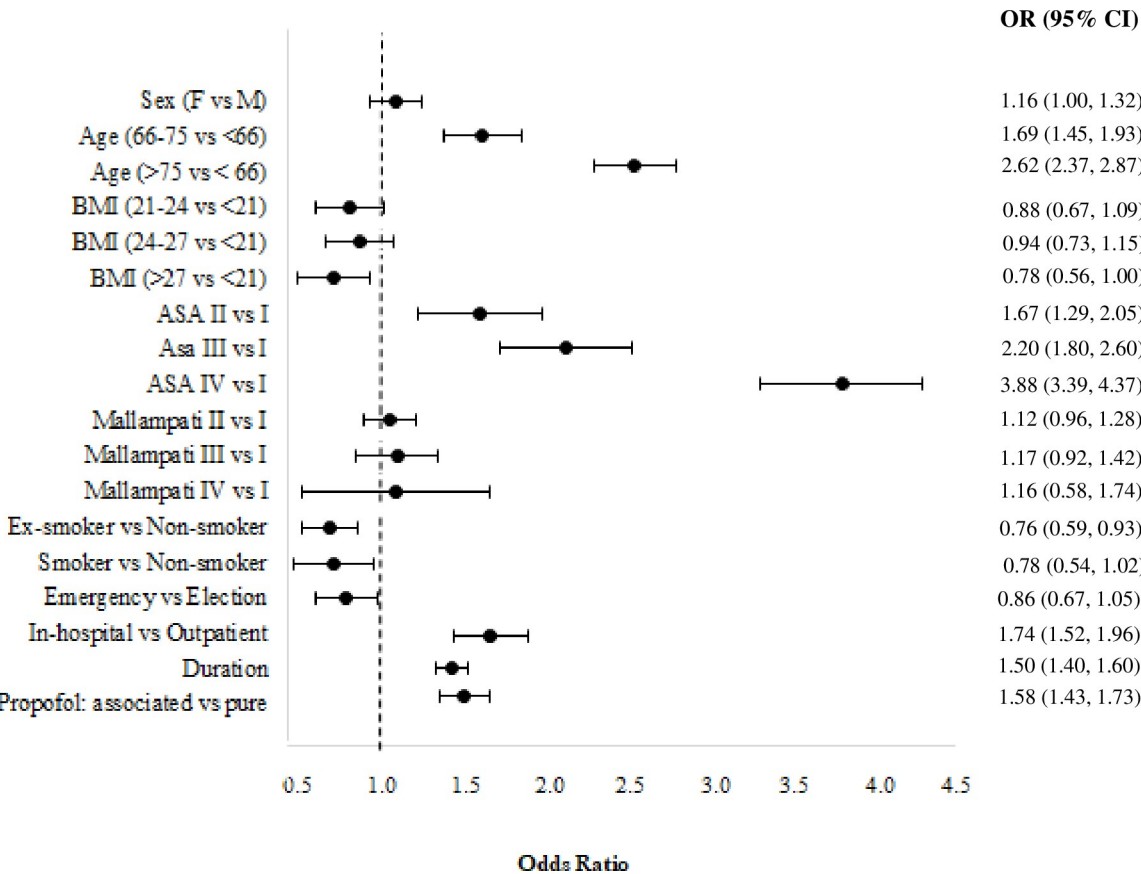

**Fig 3. Estimated odds ratios of the multinomial logit model (Model 3) for the cardiovascular adverse events, confidence intervals are provided at a confidence level of 0.95.**

of "no AE" is taken as reference. The risk of cardiovascular AEs is increased by advanced age, worse ASA score, in-hospital procedure, longer duration of the procedure, the association of midazolam/fentanyl together with propofol, and it is reduced for nonsmokers.

Similarly, Fig 4 reports the estimated odds ratios and the corresponding estimated confidence intervals for respiratory AEs. This risk is increased by higher BMI, worse ASA and Mallampati scores, emergency, in-hospital procedure, longer duration of the procedure, and the association of midazolam/fentanyl together with propofol.

Fig 5 shows the estimated parameters refereed to the hemorrhagic AEs. This risk is increased by advanced age, in-hospital procedure, longer duration of the procedure, and the association of midazolam/fentanyl together with propofol.

The risk of suspension of the endoscopic procedure before its accomplishment is not reported in figure. We notice that it is higher for females (OR 1.69), and it increases with the ASA score (OR 1.71), the Mallampati score (OR 1.48), and with in-hospital versus ambulatory procedure (OR 1.96), and it reduces by emergency with respect to the elective procedure (OR 0.42), and with duration of the procedure (OR 0.08).

## Discussion

Regarding the occurrence of AEs during sedation for GI endoscopy, our study suggests a predictive value of some covariates, such as age, BMI, ASA and Mallampati scores, in-hospital

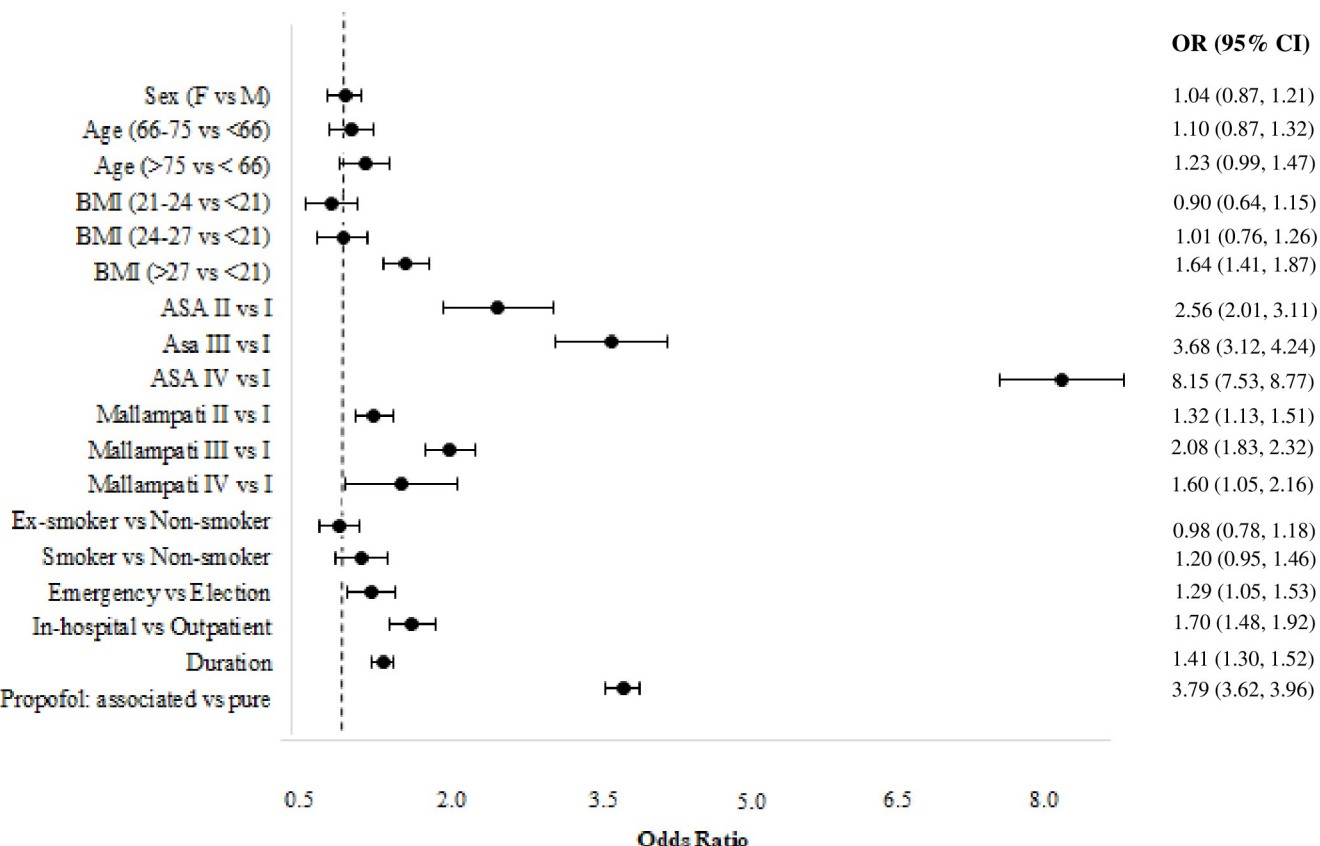

| | OR (95% CI) |
|---|---|
| Sex (F vs M) | 1.04 (0.87, 1.21) |
| Age (66-75 vs <66) | 1.10 (0.87, 1.32) |
| Age (>75 vs < 66) | 1.23 (0.99, 1.47) |
| BMI (21-24 vs <21) | 0.90 (0.64, 1.15) |
| BMI (24-27 vs <21) | 1.01 (0.76, 1.26) |
| BMI (>27 vs <21) | 1.64 (1.41, 1.87) |
| ASA II vs I | 2.56 (2.01, 3.11) |
| Asa III vs I | 3.68 (3.12, 4.24) |
| ASA IV vs I | 8.15 (7.53, 8.77) |
| Mallampati II vs I | 1.32 (1.13, 1.51) |
| Mallampati III vs I | 2.08 (1.83, 2.32) |
| Mallampati IV vs I | 1.60 (1.05, 2.16) |
| Ex-smoker vs Non-smoker | 0.98 (0.78, 1.18) |
| Smoker vs Non-smoker | 1.20 (0.95, 1.46) |
| Emergency vs Election | 1.29 (1.05, 1.53) |
| In-hospital vs Outpatient | 1.70 (1.48, 1.92) |
| Duration | 1.41 (1.30, 1.52) |
| Propofol: associated vs pure | 3.79 (3.62, 3.96) |

**Fig 4. Estimated odds ratios of the multinomial logit model (Model 3) for respiratory adverse events, confidence intervals are provided at a confidence level of 0.95.**

versus outpatient procedure, and the duration of the procedure itself. Moreover, we evidenced a significantly increased risk when different sedative drugs are administered in addition to propofol.

Such an effect of age, BMI, and ASA score is expected since more fragile patients are conceivably more prone to AEs during sedation. More prolonged endoscopic procedures not only prolong the time of exposure to the occurrence of AEs but are also frequently associated with higher operative complexity. Similarly, outpatient endoscopy tends to be performed on healthier patients and for more straightforward procedures. The predictive role on AEs of the Mallampati score, which is validated to predict tracheal intubation difficulty, is unclear, although it is confirmed by previous observations [4]: a possible role of upper airway anatomy cannot be ruled out in this setting.

Our results agree with the findings of a study previously conducted in the same hospital [4], although that study did not address the issue of multidrug sedation. In that paper, a lower incidence of AEs (4.51%) was reported compared to our present study (7.55%). This is conceivably accounted for by differences that occurred over time in the patient population. Although the mean patient age was comparable in the two settings (65 versus 64 years), the sample analyzed in Agostoni M et al. [4] had less systemic disease than patients in our present series (an ASA score of 3–4 points was registered on average for 21.96% of the patients against 31.74% observed in the sample of the current study). Moreover, a higher percentage of GI endoscopies was performed under sedation in the present cohort (33.00% versus 24.26% of the previous sample).

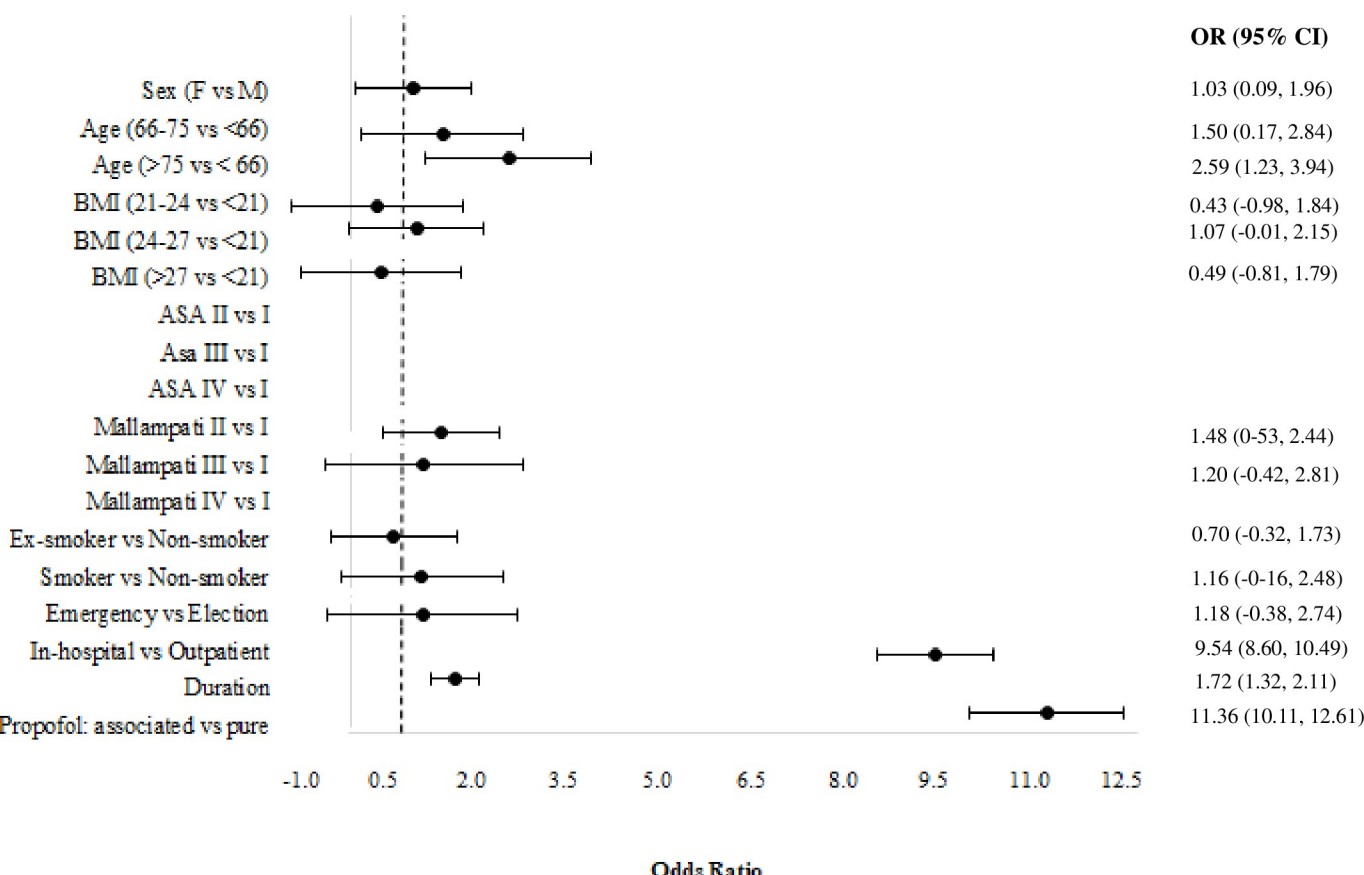

**Fig 5. Estimated odds ratios of the multinomial logit model (Model 3) for the hemorrhagic adverse events, confidence intervals are provided at a confidence level of 0.95.**

In a recent study conducted by Geng et al. [33] and based on a much smaller sample (220 cases), it was reported that hypoxemia occurring during GI endoscopy is favored by BMI, habitual snoring, and neck circumference. It is conceivable that snoring and neck circumference may be associated with a higher Mallampati score. Therefore our results agree substantially also with this report. Nevertheless, the present study's findings differ from those published by Geng et al [33] because they found no effect of age and ASA score on intraprocedural hypoxemia. These authors addressed only hypoxemia as an AE and did not consider some other important potentially predictive variables, such as in-hospital versus outpatient, the duration of the procedure, and the sedative drug association. Along with the sample size, these factors could account for the differences between their results and ours.

The adverse effect of the association of different sedatives with propofol is still a matter of debate. The Balanced Propofol Sedation (BPS) administration is reported as beneficial by Van Natta ME and Rex DK [34]. The concept of BPS is precisely the co-administration of propofol with one or more additional agents, which is supposed to reduce the total dose of propofol required, to provide analgesia (not provided by propofol alone), and to add elements of reversibility (no antidote is available for propofol, but it is for benzodiazepine and opioids). On the other hand, Van Natta ME and Rex DK [34] limit their analysis to only 200 colonoscopy patients and do not use any TCI administration device. They measured patients' satisfaction and pain and reported no difference when BPS was administered. Moreover, while their

propofol group attained deep sedation, their propofol plus opioid/benzodiazepine group attained only moderate sedation, so that comparability may be questionable.

Although the BPS rationale is theoretically acceptable, in the real-life setting of clinical sedation, it is possible that using multiple drugs with the synergistic effect would increase the difficulty of their titration. The prominent contribution of multi-drug sedation increases the risk of AEs. Since we did not measure the depth of sedation, we cannot exclude some degree of oversedation in some cases. Although the effectiveness of sedation depth monitoring has been questioned [35], this could explain why the administration of adjunctive sedatives or analgesics played a negative role in the occurrence of AEs. Nevertheless, our use of TCI warrants a correct degree of sedation in most cases [36–38].

The higher safety profile of propofol-alone sedation suggested by our study is particularly important in the setting of non-anesthesiologist sedation for GI endoscopy. This practice is purported by several guidelines [39,40], and it is implemented in several countries, although it is still debated in others [41]. An indication to avoid multi-drug sedation in the non-anesthesiologist sedation setting may be advisable in view of our results.

We did not provide any formal, strict definition of the AEs. In our opinion, this should not be regarded as a limitation of our study since we focused on clinically significant AEs that prompted the Anesthesiologist's intervention, a "real life" approach that is much more useful on practical grounds. Moreover, we did not study single AEs (such as bradycardia, desaturation), but we instead classified AEs into four major categories (cardiovascular, respiratory, hemorrhagic, and others). This approach conveys our results in a more practical way by focusing on predictive variables of AEs rather than on AEs themselves. Different GI endoscopic procedures could differently affect the occurrence of AEs. Our study, does not address this issue, and we consider only "general" characteristics of the procedures (emergency, in-hospital, duration).

A minority (1.05%) of our patients underwent general anesthesia with tracheal intubation and controlled ventilation. In these cases, anesthesia was maintained with sevoflurane; hence general anesthesia cannot be considered a confounder in comparing propofol monotherapy and propofol drug association.

A strength of our study is that data were prospectively collected, and both sedation and data records were highly homogeneous. This could be warranted, despite the natural turnover of Anesthesiologists over the study period, by the constant supervision of a single senior Anesthesiologist (co-author MA). Another important aspect of the current proposal is the methodological approach. The proposed model, in fact, adequately accounts for observations where only a few patients are at risk. This is a common occurrence when the interest is on AEs occurring during relatively safe procedures, such as sedation for GI. First of all, the analysis we employed preserves the natural structure of the data and assesses whether patients' clusters are identifiable and which group of patients is at risk. Moreover, the proposed analytical procedure suggests the most important features affecting the probability of developing AEs.

Our data confirm the previously reported effect of age, BMI, ASA score, Mallampati score, in-hospital, and procedure duration on the risk of AEs during GI endoscopy. Moreover, we suggest that the use of other sedatives together with propofol increases the risk for AEs in this setting.

## Author Contributions

**Conceptualization:** Marco Gemma, Roberta Tritto.

**Formal analysis:** Marco Gemma, Fulvia Pennoni.

**Funding acquisition:** Massimo Agostoni.

**Investigation:** Roberta Tritto, Massimo Agostoni.

**Methodology:** Fulvia Pennoni.

**Project administration:** Roberta Tritto.

**Resources:** Massimo Agostoni.

**Software:** Fulvia Pennoni.

**Supervision:** Marco Gemma.

**Writing – original draft:** Marco Gemma, Fulvia Pennoni.

**Writing – review & editing:** Marco Gemma, Fulvia Pennoni.

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
