## [Decision Letter · Decision Letter 0]

2 Oct 2020

PONE-D-20-24076

Multi-drug sedation increases the risk of adverse events along with other factors: Finite mixture model analysis of an 8-year cohort of sedations in GI endoscopy

PLOS ONE

Dear Dr. Gemma,

Thank you for submitting your manuscript to PLOS ONE. After careful consideration, we feel that it has merit but does not fully meet PLOS ONE’s publication criteria as it currently stands. Therefore, we invite you to submit a revised version of the manuscript that addresses the points raised during the review process.

Please find enclosed the detailed notes by the reviewers, who request additional information on the analyses as well as some reformatting of the manuscript.

We look forward to receiving your revised manuscript.

Kind regards,

Jörn Karhausen

Academic Editor

PLOS ONE

Journal Requirements:

2. Please include additional information regarding the study variables analysed in the study and ensure that you have provided sufficient details that others could replicate the analyses.

For instance, the American Society of Anesthesiologists (ASA) score and Mallampati scores are not described in any detail.

4. Please remove your figures from within your manuscript file, leaving only the individual TIFF/EPS image files, uploaded separately.  These will be automatically included in the reviewers’ PDF.

Reviewers' comments:

Reviewer's Responses to Questions

**Comments to the Author**

1. Is the manuscript technically sound, and do the data support the conclusions?

Reviewer #1: Partly

Reviewer #2: Partly

2. Has the statistical analysis been performed appropriately and rigorously? 

Reviewer #1: I Don't Know

Reviewer #2: No

3. Have the authors made all data underlying the findings in their manuscript fully available?

Reviewer #1: Yes

Reviewer #2: No

4. Is the manuscript presented in an intelligible fashion and written in standard English?

Reviewer #1: Yes

Reviewer #2: No

5. Review Comments to the Author

Reviewer #1: The paper concludes that multi-drug sedation increases the risk of adverse events.

1. The author should list the type and doses of sedative drugs used in they study.

2. It may be better for the authors to list baseline demographic and clinical characteristics between patients with and without multi-drug sedation.

3. They author should compare they study to previous data which showed that body mass index (BMI), habitual snoring and neck circumference were associated with hypoxemia (J Int Med Res. 2019 May;47(5):2097-2103.)

4. This study includes some patients with ASA III and IV, who may have cardiovascular and respiratory diseases before sedation. How did you differentiate it from adverse events?

Reviewer #2: The language of the paper needs to revised. For instance, We study23,788 GI endoscopiesunder sedation on consecutive adults between... What does 'consecutive' here mean??

In 1,797(7.55%) cases one or more adverse eventoccurred. Plural, events. Too many errors to note.

The abstract of the paper is written in a strange way. Especially the Conclusions part. Please revise.

The introduction of the paper is too short. It does not cover related work properly. Needs to be extended by one page.

Analysis:

The ZIPMR model is mentioned but not formally defined. Please add formulas and discuss in detail. Add also relevant references. What does the acronym mean etc?

The models are estimated through the R environment... Very strange formulation, revise.

The R code for reproducibility is available from the authors upon request. I suggest to deposit the code on github.

performed according to the better fit provided by the Bayesian information criterion.... Why? What would be alternatives?

with y being a realization of Y... Is there a subscript i missing? Why not?

The descriptions of model 1 - 3 are very brief. It should be extended by providing much more details and discussions.

The paper does not have a conclusions section. Why? Add one.

6. PLOS authors have the option to publish the peer review history of their article (what does this mean?). If published, this will include your full peer review and any attached files.

Reviewer #1: No

Reviewer #2: No

---

## [Author Response · Author response to Decision Letter 0]

6 Apr 2021

March 26, 2021

Professor Jörn Karhausen

Academic Editor, Plos One 

Dear Dr. Karhausen, 

Thank you for the referee report on October 2, 2020, of PONE-D-20-24076 titled: “Multi-drug sedation increases the risk of adverse events along with other factors: Finite mixture model analysis of an 8-year cohort of sedations in GI endoscopy”.

In the following we provide an item by item response to each comment raised by the referees.

We checked the journal style, and we changed the text accordingly. 

Note that the title of the manuscript is changed. The new title is the following: “Risk of adverse events in gastrointestinal endoscopy: Zero-inflated Poisson regression mixture model for count data and a multinomial logit model for the type of event”.

2. Please include additional information regarding the study variables analysed in the study and ensure that you have provided sufficient details that others could replicate the analyses.

For instance, the American Society of Anesthesiologists (ASA) score and Mallampati scores are not described in any detail.

Note that we expanded the description of the variables and of the available data in the Methods and the Introduction sections. We added another table to illustrate with more details the observed sample with descriptive statistics. The data are not public. However, we will add a link to our Github page with the R code, which will be made available as indicated in the paper to replicate the proposed models.

3. We note that you have indicated that data from this study are available upon request. PLOS only allows data to be available upon request if there are legal or ethical restrictions on sharing data publicly. We will update your Data Availability statement on your behalf to reflect the information you provide.

Sorry, we were wrong. The data are not publicly available.

4. Please remove your figures from within your manuscript file, leaving only the individual TIFF/EPS image files, uploaded separately. These will be automatically included in the reviewers’ PDF.

We have now removed our figures from within the manuscript and we uploaded the individual TIFF/EPS image files. 

Author Response

Reviewer #1:

The paper concludes that multi-drug sedation increases the risk of adverse events.

Please note that in the revised manuscript, all the changes are highlighted in blue. We amended the abstract, and we slightly changed the title, which now is: “Risk of adverse events in gastrointestinal endoscopy: Zero-inflated Poisson regression mixture model for count data and a multinomial logit model for the type of event”.

1. The author should list the type and doses of sedative drugs used in they study.

In the second paragraph of the Materials and Methods section, we have now added a description of the sedative drugs we used and their administration mode. 

2. It may be better for the authors to list baseline demographic and clinical characteristics between patients with and without multi-drug sedation.

We added a table (Table 2) comparing patients’ demographic and clinical characteristics according to drug sedation. We also added another table (Table 3) to show the counts with more details.

.3. They author should compare they study to previous data which showed that body mass index (BMI), habitual snoring and neck circumference were associated with hypoxemia (J Int Med Res. 2019 May;47(5):2097-2103.)

We thank the Reviewer for this suggestion since we particularly appreciated the paper by Geng W et al., and we have now added a paragraph in the Discussion section comparing their proposal with our approach and discussing their and our results, and we also mention especially a previous work carried out by some of us.

4. This study includes some patients with ASA III and IV, who may have cardiovascular and respiratory diseases before sedation. How did you differentiate it from adverse events?

Adverse events of sedation are meant to be related to sedation itself. ASA III or IV patients are more prone to experience adverse events than healthier subjects. We agree with the Reviewer that some clarification was needed here. In the Methods section, we have now better remarked that “AEs occurring during sedation were recorded if they required the anesthesiologist’s intervention”. Moreover, we have now specified that “AEs were meant to be unfavorable events related to sedation and seriously jeopardizing baseline patients’ conditions”.

Reviewer #2:

The language of the paper needs to revised. For instance, We study23,788 GI endoscopiesunder sedation on consecutive adults between... What does 'consecutive' here mean??

In 1,797(7.55%) cases one or more adverse eventoccurred. Plural, events. Too many errors to note.

Please note that in the revised manuscript, all the changes are highlighted in blue. We amended the abstract, and we slightly changed the title, which now is: “Risk of adverse events in gastrointestinal endoscopy: Zero-inflated Poisson regression mixture model for count data and a multinomial logit model for the type of event”.

We checked spelling, grammar, and punctuation, and now we provide a correct manuscript. We added much more details on the Introduction section and others in the whole manuscript, and checked the references.

The abstract of the paper is written in a strange way. Especially the Conclusions part. Please revise. 

We amended the abstract.

The introduction of the paper is too short. It does not cover related work properly. Needs to be extended by one page. 

We have somewhat expanded the Introduction. Nevertheless, please note that we covered related work in the Discussion section and our aim in the Introduction is only to provide a summary. It is currently one page long. 

Analysis: The ZIPMR model is mentioned but not formally defined. Please add formulas and discuss in detail. Add also relevant references. What does the acronym mean etc? 

Thanks for this remark. We enlarged quite a lot the methodological part of the paper. We added a detailed description of the proposed Zero-Inflated Poisson regression mixture model (ZIPMR) in the paper in the Material and Methods Section. 

The models are estimated through the R environment... Very strange formulation, revise. The R code for reproducibility is available from the authors upon request. I suggest to deposit the code on github. The R code for reproducibility is available from the authors upon request. I suggest to deposit the code on github. 

In the revised manuscript, we specify the functions used within the open-source software R, and in the manuscript, we added that the code we used would be available. We also provide the link to the page in the Github repository with the code to estimate the proposed models as suggested by the Reviewer. This file also contains some routines to perform a simple test for overdispersion as explained in the Material and Methods Section. The data are not publicly available.

performed according to the better fit provided by the Bayesian information criterion.... Why? What would be alternatives?

In the revised manuscript, we added a more detailed explanation about this information criterion; we also mention other criteria usually employed to select the best number of components with some related references added in the paper.

with y being a realization of Y... Is there a subscript i missing? Why not?

The descriptions of model 1 - 3 are very brief. It should be extended by providing much more details and discussions.

We checked and corrected all the formulas in the manuscript. Moreover, as noted above, we enlarged the paper’s methodological part and added a detailed description of the proposed the Zero-Inflated Poisson regression mixture model (ZIPMR). We also added some more details for what concerns the multinomial logit model. 

The paper does not have a conclusions section. Why? Add one.

The last paragraph of the Discussion section reports the conclusion of our paper. This is in keeping with PLOS ONE's style requirements. Nevertheless, some papers in the journal report a dedicated section for conclusions. Should this be deemed mandatory, we agree to separate the last paragraph of our Discussion under the subtitle “Conclusions”.

---

## [Decision Letter · Decision Letter 1]

8 Jun 2021

Risk of adverse events in gastrointestinal endoscopy: Zero-inflated Poisson regression mixture model for count data and multinomial logit model for the type of event

PONE-D-20-24076R1

Dear Dr. Gemma,

We’re pleased to inform you that your manuscript has been judged scientifically suitable for publication and will be formally accepted for publication once it meets all outstanding technical requirements.

Kind regards,

Jörn Karhausen

Academic Editor

PLOS ONE

Additional Editor Comments (optional):

Reviewers' comments:

Reviewer's Responses to Questions

**Comments to the Author**

1. If the authors have adequately addressed your comments raised in a previous round of review and you feel that this manuscript is now acceptable for publication, you may indicate that here to bypass the “Comments to the Author” section, enter your conflict of interest statement in the “Confidential to Editor” section, and submit your "Accept" recommendation.

Reviewer #2: All comments have been addressed

2. Is the manuscript technically sound, and do the data support the conclusions?

Reviewer #2: Yes

3. Has the statistical analysis been performed appropriately and rigorously? 

Reviewer #2: Yes

4. Have the authors made all data underlying the findings in their manuscript fully available?

Reviewer #2: No

5. Is the manuscript presented in an intelligible fashion and written in standard English?

Reviewer #2: Yes

6. Review Comments to the Author

Reviewer #2: All comments have been addressed, however, the language still needs some improvement. This will increase readability.

7. PLOS authors have the option to publish the peer review history of their article (what does this mean?). If published, this will include your full peer review and any attached files.

Reviewer #2: No

---

## [Editor Report · Acceptance letter]

14 Jun 2021

PONE-D-20-24076R1 

Risk of adverse events in gastrointestinal endoscopy: Zero-inflated Poisson regression mixture model for count data and multinomial logit model for the type of event 

Dear Dr. Gemma:

I'm pleased to inform you that your manuscript has been deemed suitable for publication in PLOS ONE. Congratulations! Your manuscript is now with our production department. 

Kind regards, 

on behalf of

Dr. Jörn Karhausen 

Academic Editor

PLOS ONE